# Recommendations from Latinx Trans and Non-Binary Individuals to Promote Cancer Prevention in Puerto Rico and Florida

**DOI:** 10.3390/ijerph20021213

**Published:** 2023-01-10

**Authors:** Joshua J. Rivera-Custodio, Ana V. Soto-Sanchez, Elvin O. Alvarado-Cardona, Fabian Moreta-Ávila, Julian Silva-Reteguis, Erik Velez-Perez, Coral Jiménez-Ricaurte, Eliut Rivera-Segarra, Sheilla L. Rodríguez-Madera, Alixida Ramos-Pibernus

**Affiliations:** 1School of Behavioral and Brain Sciences, Ponce Health Sciences University, Ponce 00732, Puerto Rico; 2Independent Researcher, Vallejo, CA 94590, USA; 3Independent Researcher, San Sebastián 00685, Puerto Rico; 4School of Public Health, Ponce Health Sciences University, Ponce 00732, Puerto Rico; 5Global and Sociocultural Studies, Florida International University, Miami, FL 33199, USA

**Keywords:** trans, non-binary, cancer prevention, Latinx

## Abstract

Latinx trans and non-binary individuals (LTNB) face increased cancer-related health disparities. Studies evidence how barriers at the individual, provider and organizational levels drive cancer disparities among LTNB individuals. These barriers increase the emotional discomfort associated with testing and disengagement from cancer prevention efforts. Moreover, there are no guidelines or interventions that address cancer prevention specifically among LTNB individuals. There is a need to develop interventions informed by the LTNB communities to promote cancer prevention and screening. The study aims to describe the recommendations provided by LTNB individuals to foster cancer screening and prevention in the communities residing in Puerto Rico and Florida. We conducted two online focus groups with a total of 15 LTNB participants. Participants were recruited using non-probabilistic purposive sampling. We used rapid-qualitative analysis for data interpretation. Findings are gathered in three main themes: (1) recommendations for promoting cancer prevention screening among providers; (2) specific recommendations to promote cancer screening among LTBN individuals; and (3) recommendations on delivery formats to foster cancer prevention. These results evidence the need and feasibility of developing community informed tailored interventions targeting cancer screening and preventative care to reduce cancer-related health disparities among the LTNB population.

## 1. Introduction

According to the American Cancer Society (ACS), cancer is the leading cause of death among the Latinx [1] population living in Continental United States and its territories [2]. Some factors that increase cancer inequities among these groups include: structural racism (i.e., discriminatory health policies), limited access to care (i.e., a lack of transportation, a lack of health insurance coverage), and cultural factors (i.e., language barriers, traditions, acculturation stress) [2,3]. Due to their intersecting identities as ethnic and gender minorities, Latinx trans and non-binary individuals (LTNB), face increased cancer-related health disparities compared to the cisgender population, such as: delays in cancer diagnosis and treatment (see Table 1 for term definitions).

Research has found that TNB (trans and non-binary individuals) were more likely to be diagnosed with advanced stage lung cancer and were less likely to receive treatment for kidney and pancreas cancers [4,5]. Furthermore, a countrywide cancer study by the National Cancer Database (NCD) showed TNB were more likely to suffer from cancers occurring in the anus, liver, skin, and lymphoma, and had lower rates of surviving non-Hodgkin lymphoma, prostate cancer, and urinary bladder cancer [5]. Additionally, TNB were less likely to receive recommended cervical, breast, and colorectal cancer screenings [6]. Studies evidencing how barriers at the individual, provider, organizational, and political levels drive these cancer inequities among TNB individuals [5,6,7]. Some individual barriers include: low cancer screening literacy; a lack of coverage for gender-affirming services; the lack of access to primary healthcare services; limited access to providers who offer gender-affirming care; and an anticipation of provider discrimination [8,9]. Provider related barriers include: a lack of provider sensitivity and cultural competence; “misgendering” or use of incorrect pronouns; stigma, discrimination, clinician attitudes and assumptions; and a lack of knowledge or formal training about TNB health, which encourages distrust of health providers [8,9]. Organizational barriers include: the exclusion of the non-binary identity in medical records; the use of binary bathrooms; and the lack of appropriate screening guidelines for TNB. Political barriers include anti-trans healthcare bills (i.e., national or state laws that penalize medical providers and TNB individuals for offering or receiving gender-affirming care) [10] (See Table 2 for PR and FL state trans healthcare laws). These barriers increase emotional discomfort associated with testing and disengagement from cancer prevention efforts [11]. Some recommendations provided by TNB to promote cancer screening include: “online scheduling, not using personal pronouns, use of gender-neutral gowns and office decor and providing all staff with basic education about the LGBTQ+ community” [9].

Moreover, there are few guidelines addressing cancer prevention among TNB individuals [13], and none are specifically tailored to LTNB. This is particularly important for engaging the Latinx communities in preventive efforts. To attend for global applicability in different cultural contexts, the Standard of Care for the Health of Transgender and Gender Diverse People, Volume 8 (SOC-8), recommends providers: offer culturally sensitive care, and involve TNB individuals in the development of health care initiatives [14]. Therefore, there is a need to develop cancer prevention and screening efforts informed by the LTNB communities to ensure the target population is reached. To address this gap, this study aims to describe the recommendations provided by LTNB individuals residing in Puerto Rico (PR) and Florida (FL) to foster cancer screening and prevention in these communities.

## 2. Materials and Methods

Using a qualitative approach, the team implemented two online focus groups with key informants using Zoom’s secure platform. Synchronous online focus groups were implemented to: (1) facilitate communication between participants living in PR and FL; and (2) follow COVID-19 risk mitigation strategies as the study was conducted during the peak of the COVID-19 pandemic. Participants were recruited using a purposive sampling. This type of sampling is based on the assumption that, given the aims of the study, specific kinds of people may hold different and important views about the ideas and issues in question and, therefore, should be purposively recruited into the sample [15]. The recruitment process began in June 2021 and culminated by September 2021, during the COVID-19 global pandemic. Recruitment was conducted in collaboration with two independent researchers’ members of the trans community (i.e., the fourth and fifth authors) who are part of the research team. The inclusion criteria for key informants were: (1) self-identify as trans, non-binary, or any other self-identifying term to represent gender diversity; (2) 21 years of age (the age of adulthood in PR); (3) identify as Latinx; and (4) currently live in PR or FL. This research is part of a more expansive NIH-funded mixed-method sequential study (1R21CA233449) examining barriers and facilitators for cervical and breast cancer screening among LTNB individuals. This study was approved by the Ponce Health Sciences University’s Institutional Review Board (IRB) Protocol: 1903009446R001).

A total of 15 participants contributed to the discussions of the focus groups: 9 participants attended the first focus group, while 6 participants attended the second focus group. Ten participants (67%) lived in PR; while five participants (33%) lived in FL. The age of participants ranged from 21–69 years. Participants received a $50 Amazon Gift Card as an incentive for participating in the focus groups. 

The purpose of the focus groups was to identify strategies for the future development of interventions to promote cervical and breast cancer prevention tailored to LTNB individuals. The research team developed a focus group guide a priori that explored topics, such: as content for intervention; strategies to encourage cervical and breast cancer screening; expectations of sensitive physicians; the preferred format of the intervention; barriers to interventions (Appendix A). Examples of focus group questions included: What topics should be included in cancer interventions tailored for gender-diverse populations? and What factors would motivate LTNB individuals to receive cancer prevention care? At the beginning of the focus groups, the moderator and primary investigator (i.e., the eighth author) asked for participants’ verbal consent, age, and place of residency. Then, the moderator presented the agenda and rules of the group discussions. Finally, the moderator offered a brief presentation with the preliminary results from the previous phases of the larger study and promoted discussion between the participants. The Research Assistants (RAs) were tasked to read comments done in the chat room and readmit participants who had logged out due to technical problems. The focus groups were audiorecorded and transcribed.

Descriptive statistics (i.e., frequencies, means) were conducted for sociodemographic data. (See Table 3). The team implemented a rapid thematic analysis--an innovative technique to obtain targeted qualitative data in a shorter period of time [16]. Following the rapid analysis process recommendations, the team developed a summary table for each focus group. The data was consolidated and divided into domains (i.e., key questions) from the summary table. The process was done by two RAs (i.e., the first and second authors). Both RAs did a separate summary of the data and discussed to ensure the consistency of findings.

## 3. Results

Findings were gathered into three main themes: (1) recommendations for the content of an intervention promoting cancer prevention; (2) specific recommendations to promote cancer screening among LTBN individuals; and (3) recommendations on the format of the potential intervention. See Table 4 for a description of these themes. Below are examples of quotations regarding a potential cancer prevention intervention. Each narrative is identified with the focus group in which they participated, their age and territory of residence.

### 3.1. Recommendations for Promoting Cancer Prevention Screening among Providers

Recommendations for promoting cancer prevention were related to improvements at the health provider level. Quotations included concerns about the education doctors have about the risks and secondary effects of hormone use in the reproductive system before beginning the medical transition and the proper management of dosage. Participants expressed the need for gender-affirming behavior and sensitivity. Specifically, participants affirmed the importance of the use of inclusive language and the correct use of pronouns by health personnel. Moreover, participants expressed the need for documentation stating the patients’ deadname and their affirmed name to avoid health personnel deadnaming or referring to patients by their non-affirmed names.

“Every clinic should have a document … specifying the [patient’s] legal name and their preferred name. The clinic’s personnel should refer to that person by their preferred name from the first moment. …”.(FG01; 30 years, PR)

Through cancer prevention efforts, participants suggested training and education for health providers regarding cancer manifestations in trans bodies of all ages. Participants perceived that a lack of trans-inclusive information on breast, cervical, uterine, or ovarian, and prostate cancer could increase the risks of these diseases and disregard screening and preventative care in LTBN individuals.

“[Doctors] should know how cervical and breast cancer looks like in transgender men, not only cisgender women. Many times, they only know what they’ve been taught in med school, which is a cisgender female body. They don’t present or inform about transgender men that can have cervical cancer and are more prone to other [diseases]. They need to know how to identify cancer and other types of diseases in non-cisgender bodies and educate their patients correctly about those differences [in disease manifestation on different bodies]”.(FG02; 21 years, PR)

Moreover, participants recommended specific training on adequate hormone dosage, its potential relationship to cancer, and the marginalization of transgender men in medical care.

“[Doctors] should be trained in [correct] hormone dosage and use of pronouns … [appropriate] hormone dosage is very important, [appropriate] pronouns are important… and visibility and treatment for transgender men too [not only transgender women]”.(FG01; 30 years, PR)

### 3.2. Specific Recommendations to Promote Cancer Screening among LTBN Individuals

Specific recommendations to promote cancer screening among LTBN individuals were targeted at the patient level. These quotes included ideas about the distribution of a gender-affirming guide for initial interviews, promotion of gender-affirmative health providers and clinics, LGBT-friendly content in medical offices (i.e., posters, stickers, flags, etc.), public policy, and mobile units. Recommendations included a buddy system during medical and legal services. Hence, emotional support through community connectedness to increase healthcare outcomes and facilitate overall well-being.

“… a type of buddy system, that’s more direct. [for example] like a call at least. Because we can’t interact in person yet [due to COVID-19 measures], so at least to have that [type] of support. To have someone that tells you ‘Okay you can do it, go and tell them this is your name, these are your pronouns, and this is the situation that is happening right now in your body’”. (FG01; 29 years, FL).A participant also suggested benefiting from community events such as PRIDE, to promote and educate about cancer prevention as it has been done with screening and prevention efforts for other diseases (i.e., HIV/AIDS).“PRIDE is an event where more than 30,000 people attend in the four days it is celebrated, statistically. I don’t see a better place to bring [education about] the topic of cancer in general. Nonetheless, for that to happen, the first step that has been discussed from the beginning: education/training for doctors … and during case management. Case management is what will initiate the dialogue with the doctor eventually, who will give a preamble of the scenario that [they] will be attending. There are resources [for that], there is training. Many [of the training] are free. [Community] organizations and hospitals have funding for that”. (FG02; 36 years, PR).

### 3.3. Recommendations on Delivery Formats to Foster Cancer Prevention

Recommendations on the format to foster cancer prevention were directed toward the community level. Participants suggested a plan to distribute brochures or material on the internet that is easy to digest and share on social networks or pages regarding trans issues. For example, a tutorial on how to do a breast self-examination that is explained by a transgender person, videos about products (i.e., binders, packers, etc.) and medications, and world news regarding trans rights.

“Written graphic material such as brochures. Videos on the internet seem like a fantastic idea, especially for preventive detection and self-examination where the breasts [are examined through] touch to see if there are any bumps or something suspicious, or how to do a prostate self-examination. That type of information could be in a video or drawn in brochures, and distributed in service centers, public plazas, or in social media or email”.(FG01; 69 years, PR)

Other quotations included ideas for mobile apps and digitized or web services with content promoted by an LTNB individual or people from their community. Participants also expressed the need for services distributed equitably between rural and urban areas and conferences about mental health awareness.

“Like a podcast or a series of interviews, YouTube videos, Facebook or Instagram lives, things like that. [Material] that is easy to share could help”.(FG02; 33 years, PR)

## 4. Discussion

In this study, the team aimed to document recommendations to improve cancer prevention strategies tailored to LTNB individuals residing in PR and FL. Consistent with previous research [17,18,19,20], these findings suggest a lack of training and competence in gender-affirming care as the main barrier for adequate cancer treatment for non-cisgender bodies.

The main finding is that LTBN individuals require culturally sensitive multilevel approaches to promote cancer prevention strategies (i.e., screenings, education) among this population. Despite the focus on cancer prevention at the service user level [18], participants indicated that promoting cancer prevention should particularly target providers and non-healthcare personnel that interact with LTNB service users in the clinics focusing on: (1) specific gender affirming behaviors (i.e., use of correct pronouns, use of the affirmed name, use of inclusive language); (2) trans care knowledge (i.e., the manifestation of cancer in trans and non-binary bodies in diverse ages, the correct dosage and risk factors of hormone use, continuous hormone use discussion in medical-patient interactions); and (3) administrative considerations (i.e., standardized sexual orientation and gender identity questions in intake with non-healthcare personnel, LGBT-friendly content in the clinic).

Findings also explored the role of public health in cancer interventions tailored to LTNB. Participants suggested the use of mobile units as a way of reducing rural health disparities. For example, some participants suggested bringing these units to specific LGBT events (i.e., Pride, other social gatherings, or events) as a way for LTNB individuals to receive early cancer screenings, regardless of geospatial limitations. These spaces of community mobilization could serve as an effective strategy to implement cancer prevention interventions and increase trust among medical professionals and researchers [21], which is essential for future collaborations with the LTNB communities.

Additionally, the team explored delivery format options for disseminating gender-affirming cancer care. According to the findings, preference varied according to age; younger participants preferred digital strategies (i.e., video, mobile apps, podcasts), while older participants preferred physical strategies (i.e., brochures, public events). Most participants agreed content should be in English and Spanish to attend cultural and ethnic diversity among LTNB individuals. While digital strategies would benefit a greater number of people, these formats could present barriers for LTNB individuals with less familiarity or access to digital content. Further studies should explore the feasibility of digital and physical gender-affirming cancer care content tailored toward LTNB individuals.

This study has several limitations. First, there was a limited representation of Latinx sub-groups, as all the participants considered themselves to be Puerto Ricans. Furthermore, the participants were LTNB individuals residing in PR and FL. As such, these results cannot be generalized for other contexts. In addition, 53% of the participants were between the 21–29 years age range. This indicates an overrepresentation of young adults in the study, and could have had an effect on the study findings. For example, there was a preference for digital delivery formats among younger participants. Additionally, sociodemographic information such as education and income were not gathered. Future studies should consider other Latinx sub-groups with diverse sociopolitical backgrounds, such as immigration status, political circumstances, education, economic status, racial identity, age, and geographic location. Other studies should consider diverse methodologies (i.e., quantitative, qualitative, mixed) to explore the multiple dimensions of cancer disparities among LTNB individuals. Future studies should also consider other contexts outside the scope of this article (i.e., LTNB individuals living in other Caribbean or Latin American countries), and increase the sample size.

In summary, despite the existence of guidelines for medical providers for engaging in cancer prevention and gender-affirming care, participants continue to emphasize this as a key priority. Targeted culturally informed interventions co-developed with the LTNB communities could be useful in addressing cancer inequities. The present article explored strategies, formats, and key target group recommendations for promoting cancer prevention among LTNB residing in PR and FL.

## 5. Conclusions

Results evidence that stakeholders have a clear picture of where and how tailored prevention efforts need to be targeted. Multi-level approaches co-developed with the LTNB communities are essential to address social, structural, cultural, organizational, and political barriers to adequate cancer care among the LTNB communities. The research team support the call to action [22] to move from an exploratory research perspective, and onwards to an intervention-based framework that reduces the health disparities of LTNB individuals. Future directions should be aimed towards: (1) the development and dissemination of LTNB sensitive affirming care guidelines for healthcare providers; (2) the development of technology-based cancer prevention interventions tailored to LTNB needs; and (3) fostering LTNB community-research partnership specifically targeting cancer prevention education.

## Figures and Tables

**Table 1 ijerph-20-01213-t001:** Terminology and definitions.

Term	Definition
Latinx	A gender-inclusive term used to refer to people of Latin American origins or descent.
Trans	Individuals whose gender identity does not match with their sex assigned at birth.
Non-binary	An individual who does not identify as strictly female or male. A non-binary person can identify as both or neither male and female, or sometimes one or the other.
Cisgender	Individuals whose gender identity matches their sex assigned at birth.
Misgender	Refers to using a word, especially a pronoun or form of address, that does not correctly reflect the gender with which a person identifies.
Deadname	Refers to a transgender or non-binary person by a name they used prior to affirming their gender identity, such as their birth name.
Gender-affirming care	A range of social, psychological, behavioral, and medical interventions that affirm diversity in gender identity and assist individuals in defining, exploring, and actualizing their gender identity, allowing for exploration without judgments or assumptions.

**Table 2 ijerph-20-01213-t002:** PR and FL state trans healthcare laws.

US Territories	State Trans Healthcare Law
Puerto Rico (PR)	Law prohibits health insurance discrimination based on sexual orientation and gender identity. State Medicaid policy explicitly covers health care related to gender transition for transgender people.
Florida (FL)	No law providing LGBTQ inclusive insurance protections. State Medicaid policy explicitly excludes transgender health coverage and care.

Note: Information of Table 2 was developed based on information gathered from the Movement Advancement Project [12].

**Table 3 ijerph-20-01213-t003:** Sample characteristics.

Variable	N	%
*Place of Residency*		
Puerto Rico	10	67%
Florida	5	33%
*Age Range*		
21–29	8	53%
30–39	4	27%
40–49	2	13%
50–69	1	7%

Note. N = 15.

**Table 4 ijerph-20-01213-t004:** Themes of results.

Theme	Description	Quotes
Recommendations for promoting cancer prevention screening among providers	Refers to key target groups for promoting cancer prevention	“I think that doctors should educate themselves more, keep evaluating and consulting cases with doctors, not wait to have a specific case and then look for information… It happened to me that I had an emergency, and basically, I’ve had to go several times to the doctor’s office because she didn’t have a correct answer for what was happening to me and had to tell me to give her time while she was looking for resources. {Doctors} should have the preparation before starting to receive patients with a trans experience so that they don’t have to ask them for this.”
Specific recommendations to promote cancer screening among LTBN individuals	Refers to strategies to promote cancer screening	“They treat you with medical curiosity; there is no respect for the person. It’s like… ‘oh wow, look at this, I have never tended a transgender person’, and they see it as a curiosity or an anomaly or something like that… they take away what is human and see you as a curiosity and that is also something that I understand happens a lot, because for them it is a thing from another world and I am trying to have treatment, treat me as a person. That’s a big part, which I understand discourages people from seeking help because… why, do they see it as an attraction? I’m a person, and I understand that that also, like… makes people not want to see doctors and that’s why… well, consequently, tests are not done, and more knowledge about health is not sought.”
Recommendations on delivery formats to foster cancer prevention	Refers to the preferred format or medium for guidelines or interventions.	“I imagine the content very user-friendly. [It should] contribute daily news, for example, this battle was won in this country, these rights have been approved… things like that. And they [should] also have tips and tricks, for example, ‘If you are on this medication, this is what you can expect,’. Little things like that.”

## Data Availability

As per our IRB protocol, the data used to develop this manuscript is not available. Any additional information can be requested by contacting the corresponding author Alixida Ramos-Pibernus at aliramos@psm.edu.

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
