# Peer review of "Recommendations from Latinx Trans and Non-Binary Individuals to Promote Cancer Prevention in Puerto Rico and Florida"

_ijerph, 2023, doi:10.3390/ijerph20021213_

Round 1
Reviewer 1 Report
The manuscript aims to engage the LTNB individuals to promote the cancer prevention, and authors had held two focus group discussion panels. The participate number is very small (15 persons), and the conclusions somehow are quite subjective, which is not suitable for publishing.
Author Response
Reviewer 1
The manuscript aims to engage the LTNB individuals to promote the cancer prevention, and authors had held two focus group discussion panels. The participate number is very small (15 persons), and the conclusions somehow are quite subjective, which is not suitable for publishing.
R: We understand the reviewer’s concern and acknowledge it is a common concern regarding qualitative research when reviewing manuscripts using a quantitative framework (Herber et al, 2020). However, although from a quantitative methods perspective 15 participants might be considered a small sample size, this is not the case when using qualitative methods such as those employed in this manuscript. Qualitative methods literature emphasizes that a sample size of 15 to 20 participants is sufficient to achieve data saturation (Saumure & Given, 2008). Furthermore, other articles published in this journal with similar populations (i.e., latinx trans and non-binary folks) and using qualitative methodologies (i.e., focus groups) have used sample sizes of 15 participants (Castellon-Lopez et al., 2022).
Reviewer 2 Report
The work presented by Rivera-Custodio et al. highlights the high cancer incidence in the LTNB community and the need for health providers to improve care and cancer prevention and treatment strategies. The paper is well-written and easy to follow, and also results relevant to the field, however, some questions arise:
Major
1. As was mentioned before, the authors present relevant recommendations about health for the LTBN community, however, the needs and some differences in the healthcare policies can vary from one state to the other, then a small sample can be more specific other than general for the LTBN community in the US. Florida and Puerto Rico are two places with a high presence of Latin communities, however other states with large Latino populations such as California and Texas were not included in the study which represents a limitation of the presented research as the authors indicated. I suggest changing the title and the objective to the population from the two studied locations and expanding the sample.
2. The sampling could be improved by the use of a validated questionnaire; this means that it has been developed to be administrated among the study participants. This will increase the adequacy reliability and validity. This would first test the healthcare satisfaction of the participants and then collect recommendations.
3. Cancer screening recommendations considering the genetic background and the cancer incidence among Latin populations could be a good complement to the study. Then, including health providers in the discussion groups could be beneficial.
Minor
1. DOI is missing for reference 6
Author Response
The work presented by Rivera-Custodio et al. highlights the high cancer incidence in the LTNB community and the need for health providers to improve care and cancer prevention and treatment strategies. The paper is well-written and easy to follow, and also results relevant to the field, however, some questions arise:
Major
- As was mentioned before, the authors present relevant recommendations about health for the LTBN community, however, the needs and some differences in the healthcare policies can vary from one state to the other, then a small sample can be more specific other than general for the LTBN community in the US. Florida and Puerto Rico are two places with a high presence of Latin communities, however other states with large Latino populations such as California and Texas were not included in the study which represents a limitation of the presented research as the authors indicated. I suggest changing the title and the objective to the population from the two studied locations and expanding the sample.
R: We appreciate the reviewer comment and agree with the noted variation of healthcare policies per state. Thus, we have now changed the title to: “Recommendations from Latinx trans and non-binary individuals to promote cancer prevention in Puerto Rico and Florida”. Additionally, we have also modified the study’s objective in lines 92-95: “To address this gap, this study aims to describe the recommendations provided by LTNB individuals residing in Puerto Rico (PR) and Florida (FL) to foster cancer screening and prevention in these communities”
- The sampling could be improved by the use of a validated questionnaire; this means that it has been developed to be administrated among the study participants. This will increase the adequacy reliability and validity. This would first test the healthcare satisfaction of the participants and then collect recommendations.
R: We understand the reviewer’s concern and acknowledge it is a common concern regarding qualitative research when reviewing manuscripts using a quantitative framework (Herber et al, 2020). However, because this is a qualitative study with a focus group approach, we did not employ the use of questionnaires as a data collection strategy. The objective of this study was to better understand LTNB individuals’ subjective experiences regarding cancer screening and to describe their recommendations to foster cancer screening and prevention in their communities. For this reason, we did not assess healthcare satisfaction in this study. However, we will assess healthcare satisfaction in a future quantitative study.
- Cancer screening recommendations considering the genetic background and the cancer incidence among Latin populations could be a good complement to the study. Then, including health providers in the discussion groups could be beneficial.
R: We appreciate the reviewer’s observation and will consider including genetic background in future studies.
Minor
- DOI is missing for reference 6
R: This reference does not have DOI. However, we have now added the PMID and PMCID to the reference for it to be identified in the web.
Reviewer 3 Report
In the manuscript entitled " Recommendations from Latinx trans and non-binary individuals to promote cancer prevention. The authors shed light on recommendations on cancer awareness for LTNB community.
Minor comments:
1) Line 123 two hyphens can be removed.
Major comments:
1) Is there any specific difference between transgender community and LTNB community, does LTBN community has a higher rate of cancer than the trans in general?
2) The number of LTNB population taken for this study was 15, does this have any statistical significance, that too when they are from different countries and their social acceptance, economical difference, and awareness in the communities are markedly different?
3) Are there any state law or federal laws to protect the trans community?
4) Section 3.2: Listing hospitals or doctors who are LTNB/trans-friendly would be good idea.
5) What are the economic and educational differences between these 15 individuals, their respective educational/economical difference could also affect their confidence in dealing with the problem with confidence?
6) Section 3.3: conducting advertisements via posters or sessions in transgender/LTNB bars, pick-up bars, and clubs would help spread awareness.
Author Response
In the manuscript entitled " Recommendations from Latinx trans and non-binary individuals to promote cancer prevention. The authors shed light on recommendations on cancer awareness for LTNB community.
Minor comments:
1) Line 123 two hyphens can be removed.
R: We have now removed the hyphens in the following words in line 123: “re-admit” and “audio-recorded”.
Major comments:
1) Is there any specific difference between transgender community and LTNB community, does LTBN community has a higher rate of cancer than the trans in general?
R: We appreciate the reviewer’s observation and opportunity to expand on this. The short response is yes, there are differences. Specific answer below:
- We use the broad term “Latinx trans and non-binary" because it encompasses the diversity of gender identity (a person's sense of their gender) in our study. The transgender term as an umbrella, includes those participants whose gender identity does not align with their assigned sex at birth (I.e., a person assigned male at birth identifying as female). “Non-binary" is used to describe people who do not identify as strictly female or male. A non-binary person can identify as both or neither male and female, or sometimes one or the other.
- Mortality rates among Latinx individuals is higher than the general population (lines 39-40) and trans and non-binary individuals, face even more cancer-related health disparities compared to the cisgender population (line 44-47).
2) The number of LTNB population taken for this study was 15, does this have any statistical significance, that too when they are from different countries and their social acceptance, economical difference, and awareness in the communities are markedly different?
R: We understand the reviewer’s concern and acknowledge it is a common concern regarding qualitative research when reviewing manuscripts using a quantitative framework (Herber et al, 2020). However, it is important to clarify that statistical significance is not the objective of qualitative studies such as this one.
3) Are there any state law or federal laws to protect the trans community?
R: We included Table 2 that briefly presents some State Law Healthcare Laws. Furthermore, we have included a citation about political barriers to adequate gender affirming care in line 67-70: “Political barriers include anti-trans healthcare bills (i.e., national or state laws that penalize medical providers and TNB individuals for offering or receiving gender-affirming care) [10] (See Table 2 for PR and FL state trans healthcare laws)”.
- Section 3.2: Listing hospitals or doctors who are LTNB/trans-friendly would be good idea.
R: We appreciate the reviewer’s concern. Unfortunately, although we might personally agree, this was not mentioned by the study participants and thus we cannot not include it as part of the findings of this study.
5) What are the economic and educational differences between these 15 individuals, their respective educational/economical difference could also affect their confidence in dealing with the problem with confidence?
R: We appreciate the reviewer observation and agree this might be important information to better interpret our findings. We have now included it as part of the limitations of the study and hope to collect this information in future studies. See line 305: “Also, sociodemographic information such as education and income were not gathered”.
6) Section 3.3: conducting advertisements via posters or sessions in transgender/LTNB bars, pick-up bars, and clubs would help spread awareness.
R: In Section 3.3 we present the results of the focus groups (recommendations given by the participants). We included the reference of social gatherings as a recommended place for cancer prevention interventions in lines 274-276: “For example, some participants suggested bringing these units to specific LGBT events (i.e., Pride, other social gatherings or events) as a way for LTNB individuals to receive early cancer screenings--regardless of geospatial limitations”.
Reviewer 4 Report
Overall, the quality of the manuscript is fine. I would like to offer some comments as follows:
1. In the introduction, the author's current research gap presentation is not very clear.
2. Can demographic characteristics provide more details?
3. the survey population is overrepresented by 21-29 year old, does this have no effect on the study results?
4. I feel that the suggestions of 3.1 and 3.2 can be placed in section 4.2 "Practice Implications". The following structure is suggested 4 Discussion 4.1 Key findings 4.2 "Practical contributions"
5. It is recommended to conclude with a brief explanation of the shortcomings of this study and future research directions
Author Response
Overall, the quality of the manuscript is fine. I would like to offer some comments as follows:
- In the introduction, the author's current research gap presentation is not very clear.
R: We appreciate the reviewer’s observation. We have now modified lines 91-95 to introduce more clearly the research gap following the study aim: “Therefore, there is a need to develop cancer prevention and screening efforts informed by the LTNB communities to ensure the target population is reached. To address this gap, this study aims to describe the recommendations provided by LTNB individuals residing in Puerto Rico (PR) and Florida (FL) to foster cancer screening and prevention in these communities.
- Can demographic characteristics provide more details?
R: We agree with the reviewer’s observation. Unfortunately, no more sociodemographic information was collected for the study. We have now listed this as part of the study’s limitations.
- the survey population is overrepresented by 21-29 years old, does this have no effect on the study results?
R: We appreciate the reviewer’s observation on the potential effects of overrepresentation based on age. We have now added a sentence (lines 269-270) as part of the limitations of our study, for future research to consider: “In addition, 53% of the participants were between the 21-29 years age range. This indicates an overrepresentation of young adults in the study and could have had an effect on the study findings. For example, there was a preference for digital delivery formats among younger participants.”.
- I feel that the suggestions of 3.1 and 3.2 can be placed in section 4.2 "Practice Implications". The following structure is suggested 4 Discussion 4.1 Key findings 4.2 "Practical contributions"
R: We appreciate this recommendation. However, the quotes in Sections 3.1 and 3.2 are part of the Results Section. We rearranged the discussion section to make the key findings more straightforward.
- It is recommended to conclude with a brief explanation of the shortcomings of this study and future research directions
R: We greatly appreciate the reviewer’s observations. We have now included a brief summary section in lines 313-318: “In summary, despite the existence of guidelines for medical providers for engaging in cancer prevention and gender-affirming care, participants continue to emphasize this as a key priority. Targeted culturally informed interventions co-developed with the LTNB communities could be useful in addressing cancer inequities. The present article explored strategies, formats, and key target group recommendations for promoting cancer prevention among LTNB residing in PR and FL.
The study limitations are now in lines 296-312: “This study has several limitations. First, there was a limited representation of Latinx sub-groups, as all the participants considered themselves to be Puerto Ricans. Furthermore, the participants were LTNB individuals residing in PR and FL. As such, these results cannot be generalized for other contexts. In addition, 53% of the participants were between the 21-29 years age range. This indicates an overrepresentation of young adults in the study and could have had an effect on the study findings. For example, there was a preference for digital delivery formats among younger participants. Also, sociodemographic information such as education and income were not gathered. Future studies should consider other Latinx sub-groups with diverse sociopolitical backgrounds, such as immigration status, political circumstances, education, economic status, racial identity, age, and geographic location. Other studies should consider diverse methodologies (i.e. quantitative, qualitative, mixed) to explore the multiple dimensions of cancer disparities among LTNB individuals. Future studies should also consider other contexts outside the scope of this article (i.e., LTNB individuals living in other Caribbean or Latin American countries), and increase the sample size.
Finally, future directions are included in lines 327-331: “Future directions should be aimed towards: 1) the development and dissemination of LTNB sensitive affirming care guidelines for healthcare providers; 2) the development of technology-based cancer prevention interventions tailored to LTNB needs; and 3) fostering LTNB community-research partnership specifically targeting cancer prevention education.”
Round 2
Reviewer 2 Report
the comments were properly addressed
Reviewer 4 Report
The author made changes to the comments I made. The author also gives a reasonable explanation for what the author cannot correct. Therefore, I agree to accept it.